# Recent Evolution of Susceptibility to Beta-Lactams in *Neisseria meningitidis*

**DOI:** 10.3390/antibiotics12060992

**Published:** 2023-06-01

**Authors:** Ala-Eddine Deghmane, Eva Hong, Muhamed-Kheir Taha

**Affiliations:** Invasive Bacterial Infections Unit, Institut Pasteur, Université Paris Cité, 75724 Paris, France; eva.hong@pasteur.fr (E.H.); muhamed-kheir.taha@pasteur.fr (M.-K.T.)

**Keywords:** *Neisseria meningitidis*, beta-lactams, antimicrobial resistance, *penA* gene sequence, three-dimensional structure

## Abstract

Beta-lactams are the main antibiotics for the treatment of invasive meningococcal disease. However, reduced susceptibility to penicillin G is increasingly reported in *Neisseria meningitidis* and reduced susceptibility to third-generation cephalosporines (3GC) and the rare acquisition of ROB-1 beta-lactamase were also described. Modifications of penicillin-binding protein 2 (PBP2) encoded by the *penA* gene are the main described mechanism for the reduced susceptibility to penicillin and to other beta-lactams. *penA* modifications were analyzed using the sequences of all *penA* genes from cultured isolates between 2017–2021 in France (*n* = 1255). Data showed an increasing trend of reduced susceptibility to penicillin from 36% in 2017 to 58% in 2021. Reduced susceptibility to 3GC remained limited at 2.4%. We identified 74 different *penA* alleles and *penA1* was the most frequent wild-type allele and represented 29% of all alleles while *penA9* was the most frequently altered allele and represented 17% of all alleles. Reduced susceptibility to 3GC was associated with the *penA327* allele. The amino acid sequences of wild-type and altered PBP2 were modeled. The critical amino acid substitutions were shown to change access to the active S310 residue and hence hinder the binding of beta-lactams to the active site of PBP2.

## 1. Introduction

*Neisseria meningitidis* is the agent of invasive meningococcal disease (IMD). This bacterium is usually carried asymptomatically in the nasopharynx in about 10% of the general population [1] but can be responsible for devastating invasive infections with an overall notification rate in Europe of 0.9 per 100,000 that varies between 0.3 and 2.9 [2]. IMD patients require hospitalization and prompt management using antibiotics that should be administered even prior to admission to hospital if septic shock is suspected [3]. The patient should be transferred urgently to the hospital, giving preference to facilities with a resuscitation service adapted to the age of the patient. The antibiotic treatment relies mainly on the use of beta-lactam antibiotics [4,5]. The choice of these antibiotics considers their activity on *N. meningitidis*, their pharmacology, their safety, the literature, and clinical experience.

Standardization of the technical conditions for antimicrobial susceptibility testing during the early 2000s allowed reliable testing and comparison of large collections of meningococcal isolates within the European Monitoring Group on Meningococci (EMGM) member countries [6]. Later, these studies showed that invasive meningococcal isolates remained susceptible to beta-lactam antibiotics with no resistance reported in Europe. However, 38% of isolates had reduced (or intermediate) susceptibility to penicillin G, PenI or penicillin intermediate, (minimal inhibitory concentration, MIC ≥ 0.125 mg/L and ≤1 mg/L) but with variable proportions between countries [7,8]. The PenI phenotype was correlated with changes in the sequence of the penicillin-binding protein 2 (PBP2) encoded by the chromosomal *penA* gene [9]. PBP2 is a transpeptidase that ensures cross-linking of peptides through binding to the terminal dipeptide, acyl-D-alanyl-D-alanine (D, D transpeptidase), that is attached to the saccharide units of the peptidoglycan [7]. The distal D-ala is also released, leaving the debated assumption that PBP2 may be also a D, D carboxypeptidase [7].

Changes in five critical residues of PBP2 (F504, A510, I515, H541, and I566) were shown to be directly associated with the PenI phenotype and are located near the active site of the transpeptidase moiety of PBP2 [7]. The penicillin-binding site on PBP2 harbors three conserved motifs located in meningococcal PBP2 as follows: S310xxK313 (that has the reactive S310), the second motif S362xN364, and the third motif K497TG499. These motifs fill the active site cleft of PBP2. Modifications in the five residues mentioned above were suggested to modify transpeptidase activity, leading to lower transpeptidase activity with the accumulation of the pentapeptide precursor in peptidoglycan and a lower affinity of penicillin binding to PBP2 [7,10]. Indeed, PBP2 is inhibited by beta-lactams that resemble the substrate of PBP2, the terminal dipeptide, and form a covalent bond with the active site serine residue (S310) [7,11].

In recent years, several observations were published reporting the increase in the proportions of PenI isolates across the world [12]. More recently, isolates with reduced susceptibility to third-generation cephalosporins were detected in France and other countries [8,13] in addition to beta-lactamase-producing meningococci [14,15,16,17]. Moreover, the epidemiological changes following the emergence of the COVID-19 pandemic were recorded with a drastic decrease in the number of cases of IMD [18]. However, no data are yet available on any changes in the susceptibility to beta-lactam antibiotics. We aimed in this study to record the evolution of meningococcal susceptibility to these antibiotics in France during the five-year period 2017–2021, before (2017–2019), and since the emergence of COVID-19 (2020–2021). We also aimed to explore the impact of the changes in *penA* on the PBP2 structure and its binding to penicillin.

## 2. Results

Our laboratory harbors the National Reference Centre for Meningococci and *Haemophilus influenzae* (NRCMHi). Under our mission of epidemiological surveillance, we received and characterized a total of 1595 biologically confirmed IMD cases for the period 2017 to 2021. Most of these cases were confirmed by PCR and/or culture (*n* = 1255; 79%) while the remaining cases were PCR-confirmed only (*n* = 340; 21%) and for which no cultured isolates were available. The cases were mainly from the 3-year period 2017–2019 (*n* = 474, *n* = 397, *n* = 416 respectively), while the years 2020 and 2021 accounted for 202 and 106 cases, respectively. We have already reported serogroup distribution for this period with 791 serogroup B cases (49.6%), 286 serogroup C cases (17.9%), 271 serogroup W cases (17%), 221 serogroup Y cases (13.9%), and 26 cases of other groups and non-groupable cases (1.6%) [19]. In this work, we focused on the 1255 cultured isolates for antibiotic susceptibility testing.

### 2.1. Beta-Lactam Susceptibility Phenotypes

MIC ranged between 0.004 and 0.75 mg/L. MIC50 and MIC90 were 0.094 and 0.38 mg/L respectively. MIC values showed an increasing trend over the 5-year period of the study with an overall 576 isolates (46%) with reduced susceptibility to penicillin G (MIC ≥ 0.125 mg/L but MIC ≤ 1 mg/L) [7], with the highest values during the year 2021 with 58% of PenI isolates. The proportion of PenI isolates during the period 2017–2019 was 43% and increased to 57% during the period 2020–2021 (*p* = 0.0004). During the period 2017–2021, there were 30 isolates (2.4% of cultured isolates and were mainly detected during 2017–2019) with reduced susceptibility to 3GC (MIC of cefotaxim ranging between 0.047–0.125 mg/L). Almost all these isolates belonged to serogroup C. These isolates also showed reduced susceptibility to penicillin G.

Group B isolates of the PenI phenotype had their proportions increased from 58% of all group B isolates in 2017 to 76% in 2021. PenI group B isolates also accounted for 61% of all PenI isolates, increasing to 81% in 2021. One isolate of group Y was beta-lactamase-positive ROB-1 type and showed MIC of penicillin G of 3 mg/L. Table 1 shows the most frequent penA alleles that were represented by more than five isolates.

### 2.2. Distribution of penA Alleles

As modifications of PBP2, encoded by the penA gene, were shown to be correlated with the PenI phenotype [9,20], we therefore sequenced cultured isolates extracted from the whole genome data, penA alleles, and genotypes. We obtained penA sequence data from 1245 isolates out of the 1255 cultured isolates (99%).

We detected 74 penA alleles that were represented by one isolate (0.1%) to 365 isolates (29%). The most frequent penA allele was penA1 (*n* = 365, 29%). Twenty-four penA alleles were represented by at least five isolates accounting for 94% of all cultured isolates with penA sequence data. These alleles belonged to the two previously identified groups of penA alleles based on the presence/absence of the modification at the five critical positions of PBP2 (F504, A510, I515, H541, and I566). Seven alleles (1, 2, 3, 5, 22, 27, and 34) belonged to the penA^S^ group that harbors alleles encoding wild-type PBP2. These alleles accounted for 54% of the isolates of this study with penA sequence data (Table 1). Seventeen penA alleles belonged to the penA^I^ group encoding modified PBP2s with mutations in the five critical residues mentioned above. This group accounted for 40% of the isolates with penA sequence data. The remaining 6% of penA alleles were represented by less than four isolates. It is worthwhile to note that all isolates with reduced susceptibility to 3GC harbored the same modified penA327 allele.

There was an excellent correlation between the groups of penA alleles (penA^S^ and penA^I^) and the MIC of penicillin G. Isolates harboring each of the seven penA^S^ alleles all showed geometric means (GM) of MICs lower than 0.125 mg/L, with the upper limit of a 95% confidence interval lower than 0.125 mg/L. In contrast, isolates harboring the 17 penA^I^ alleles all showed GM values of equal or higher than 0.125 mg/L (Table 1).

The detection of penA^I^ alleles allowed the prediction of the PenI phenotype with sensitivity, specificity, negative predictive value, positive predictive value, and accuracy higher than 94% (Table 2). The kappa coefficient between penA prediction and phenotypic determination of MIC was nearly perfect (0.91).

### 2.3. Structure-Function Analysis of Wild-Type and Modified PBP2

We next explored the impact of the mutations in the five critical residues on the structure and function of PBP2. Figure 1A,B shows the predicted structure of PBP2 encoded by the most frequent allele (penA1), showing the active site cleft. In this model, the active site is composed of α-helix-beta sheet structure that includes the SxxK motif facing another α-helix that contains the SxN motif and the KTG motif that lies within a beta-sheet at the bottom of the active site. The S310 and K313 of the first conserved motif as well as the S362 in the second conserved motif are involved in proton transfer with the D-alanine of the donor chain in peptidoglycan biosynthesis in order to form the bond during the transpeptidase reaction. This role is also suggested for these residues when binding penicillin G. The third motif (KTG) participates by forming a hydrogen bond with the substrate [21]. Figure 1C shows the superposition of the previous structure of the wild-type PBP2 with that encoded by penA9, the most frequently modified penA^I^ allele. Overall, the PBP2 structure encoded by penA9 overlaps closely with the wild-type PBP2 structure, but the data suggest that modifications in the five critical residues (F504, A510, I515, H541, and I566) can lead to an important structural modification in the active site cleft (Figure 1C). In particular, the residues K338-R345 are structured differently between the two proteins where this region seems to bend inside the active cleft in the PBP2 encoded by penA9, compared to its extended conformation in the wild-type PBP2. This change brings the P341 residue inside the cleft and hence perturbs the interaction of the active S310 with the substrate (Figure 1C).

These data strongly suggest that the modification in penA^I^ alleles leads to important structural changes of the corresponding PBP2 responsible for the PenI phenotype through less efficient interaction of PBP2 with penicillin G.

## 3. Discussion

The use of *penA* sequencing was already suggested as a molecular tool to identify meningococcal isolates with reduced susceptibility to penicillin G based on the modification in five critical residues [7]. In that work, 139 *penA* alleles were identified and allowed this genotype/phenotype correlation. There are currently 1188 *penA* alleles in the PUBMLST database (accessed on 22 November 2022) with variable frequencies. Our study detected 74 alleles in France in the period 2017–2021. Our data still supported the use of *penA* sequencing as an excellent correlation was detected between the GM of the MIC of penicillin G and the presence/absence of the modification at the critical positions of PBP2 (F504, A510, I515, H541, and I566). This use will be helpful and informative in particular for PCR-confirmed cases where conventional antibiotic susceptibility testing by antibiogram cannot be performed. Similar characterization of meningococcal isolates with similar conclusions on the PenI phenotype and the mosaic structure of the *penA* gene were reported in the USA for the period 2012–2016 [22]. Data from England, Wales, and Northern Ireland from 2010/11 to 2018/19 were also similar to our work. Non-susceptibility to penicillin G was reported to vary during that period from 29.7% to 47.8%, in addition to several other agreements with our study [23]. The *penA1* allele was the most frequent among penicillin G-susceptible isolates and *penA9* was the most frequent allele among non-susceptible isolates. Moreover, seven isolates had cefotaxime MICs ≥ 0.047 mg/L, including one resistant isolate (MIC = 0.25 mg/L). Most of these isolates harbored the *penA327* allele [23].

The second major result of our work is the structure-function analysis that suggests a mechanism of action for the critical modifications. Our analysis took advantage of the presence of several established structures of equivalent PBP in other bacterial species. This analysis considers the conserved structure (helices and sheets of the active sites). The *penA* gene in *N. meningitidis* encodes an essential protein, PBP2, whose functions are required for bacterial viability [7]. The five critical residues do not reside within the active site. Indeed, our structure model clearly suggests that the *penA9* allele (the most frequently modified *penA^I^* allele) encoded a PBP2 protein that did not reveal major changes within the active site that were superimposed on that of the wild-type PBP2 (Figure 1). Our modeling of the PBP2 structure suggests that modifications in the critical five residues may provoke the protrusion of an unstructured loop (K338-R345) in the cleft of the active site that may reduce the attack activity of the OH group of the residue S310 of the active site [7,24]. This activity is responsible for the covalent bond of the penicillin to the active S310 of PBP2 [7], as the beta-lactam ring is similar to the D-Ala-D-Ala dipeptide of the peptidoglycan (pentapeptide precursors). This may suggest that modification in *penA9*-encoded PBP2 may lead to reduced carboxypeptidase activity that removes the last D-Ala residue during peptidoglycan biosynthesis. This hypothesis supports our previous suggestion that PBP2 may also act as a D,D carboxypeptidase as reflected by the increase in pentapeptides containing precursors during peptidoglycan biosynthesis [7]. Indeed, PBP2 is a class B PBP that usually catalyzes the D, D-transpeptidation of peptidoglycan. However, we did not detect any reduction in the peptidoglycan cross-linking degree in PenI isolates compared to Pen^S^ isolates [7]. The preservation of the transpeptidase activity in PenI mutants could be explained by the larger size of the peptidoglycan compared to beta-lactams. The various contacts expected between PBP2 and the peptidoglycan (the peptide and the glycan strand) may help to circumvent the barrier generated by the reduced susceptibility mutations, preserving therefore sufficient transpeptidase activity to support cell growth. In the absence of structural information for the interaction of PBP2 with its peptidoglycan substrate, the discrimination remains unclear. Our work provides support to the hypothesis that meningococcal PBP2 may also act as a D, D carboxypeptidase. Alternatively, PBP2 may impact the carboxypeptidase mediated by another meningococcal PBP such as PBP3 that may interact with PBP2. The alteration of PBP2 can then modify this interaction and result in the reduction of carboxypeptidase activity.

## 4. Materials and Methods

### 4.1. Bacterial Methods

Bacterial isolates are received at the NRCMHi as part of its mission of epidemiological surveillance, which includes the surveillance of susceptibility and resistance to antibiotics. Meningococcal isolates are cultured on GCB medium with Kellogg supplements [25]. Antimicrobial susceptibility testing was performed using MHF agar plates (BioRad, Marnes-la-Coquette, France) and Etest strips (BioMérieux, Marcy l’Etoile, France). The tests were performed through swabbing of an inoculum of 0.5 MacFarland turbidity on MHF plates as previously described [6]. Beta-lactams tested were penicillin G, amoxicillin, and third-generation cephalosporins (cefotaxim). Breakpoints were according to the EMGM recommendations and, in particular, reduced susceptibility to penicillin G was defined as MIC ≥ 0.125 mg/L but ≤1 mg/L [6].

### 4.2. DNA Sequencing

All cultured isolates were submitted to whole genome sequencing by Illumina technology (NextSeq 500, Illumina, Every, France) with paired-end strands of 150 bp and a sequencing depth of 50X as previously described [19]. *penA* alleles were extracted using the PUBMLST databases that provide the allele numbers as well as the presence or absence of the critical modifications at residues F504, A510, I515, H541, and I566 of PBP2 [26].

### 4.3. Structure-Function Analysis

The whole sequences of *penA* genes were translated for *penA1* and *penA9* alleles and loaded onto Phyre2 databases “www.sbg.bio.ic.ac.uk/phyre2/ (accessed on 23 September 2022)” that used known structures of similar proteins to model the structure of submitted proteins using Protein Homology/analogY Recognition Engine V 2.0. When building the structure, the modeling considers the conserved helices and sheets but not the type of the residue, and only the spatial proximities of residues are scored with iterations of refitting the structures.

The different models were thereafter analyzed and visualized using Chimera software developed by the University of California at San Francisco [27].

## 5. Limitations and Future Implications

One major limitation of our study is that it only analyzed invasive isolates, as no carriage isolates were included. However, these latter isolates may serve as a reservoir for beta-lactam resistance. However, our study provides additional phenotypical data that are linked to the characterization of additional *penA* alleles that should improve linking phenotypes to genotypes analysis through the Neisseria MultiLocus Sequence Typing website “https://pubmlst.org/ (accessed on 22 November 2022)”. Structure-function characterization may help the definition of new beta-lactams.

## 6. Conclusions

The PenI phenotype is increasingly described. This work provides further support to the link of the PenI phenotype to the modification in critical residues in PBP2 encoded by altered *penA* alleles and the use of *penA* sequencing as a tool to detect reduced susceptibility to penicillin G in meningococci. This work also provides a structure-function analysis of the consequences resulting from the modification of PBP2 that can help in designing active beta-lactams on PenI isolates.

## Figures and Tables

**Figure 1 antibiotics-12-00992-f001:**
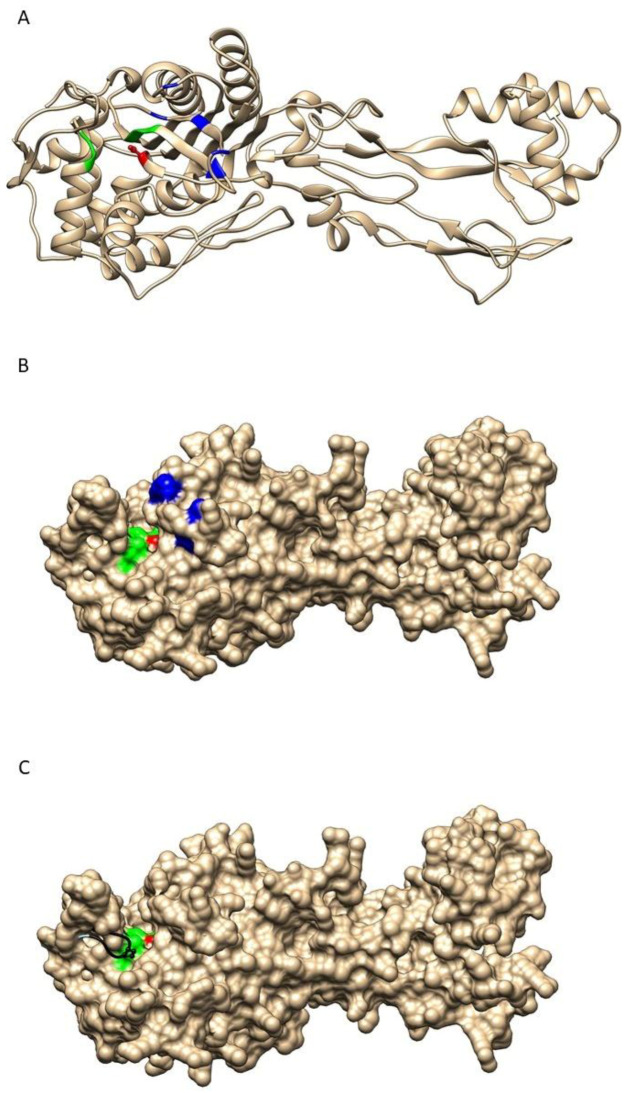
Representation of modeled PBP2 structure. (**A**) Ribbon representation of the wild-type PBP2. The active site is shown in color with the active S310 in red as balls and sticks. SxN and KTG motifs are shown in green while the five critical residues (F504, A510, I515, H541, and I566) are in blue. (**B**) is a surface view of the structure in (**A**). (**C**) A representation of the surface structure of wild-type PBP2 with the active S310 in red and the SxN and KTG motifs in green. The modeled structure of PBP2 encoded by *penA9* is superimposed on that of the wild type. The difference between the two backbones (wild-type PBP2 and modified PBP2 are shown for the modified PBP2 encoded by *penA9* with the deviation produced colored in black as a ribbon structure showing the P341 in balls and sticks.

**Table 1 antibiotics-12-00992-t001:** The most frequent *penA* alleles according to their penicillin G MIC and PBP2 alterations. 95% CI, 95% confidence intervals; GM, geometric mean.

*penA* Allele	N° of Isolates (%)	Presence of Mutation in the 5 Critical Residues	GM of Penicillin G MIC (95% CI)
1	365 (29.3)	no	0.05 (0.05–0.05)
9	208 (17.7)	yes	0.22 (0.21–0.24)
22	148 (11.9)	no	0.06 (0.05–0.07)
2	69 (5.5)	no	0.06 (0.05–0.06)
386	52 (4.2)	yes	0.14 (0.13–0.16)
3	46 (3.7)	no	0.05 (0.05–0.06)
14	42 (3.4)	yes	0.17 (0.15–0.19)
33	39 (3.1)	yes	0.22 (0.19–0.26)
327	30 (2.4)	yes	0.43 (0.38–0.49)
7	18 (1.4)	yes	0.19 (0.16–0.23)
13	17 (1.4)	yes	0.33 (0.29–0.37)
5	14 (1.1)	no	0.05 (0.04–0.06)
34	14 (1.1)	no	0.05 (0.04–0.07)
12	13 (1.0)	yes	0.20 (0.18–0.23)
19	13 (1.0)	yes	0.21 (0.15–0.3)
27	12 (1.0)	no	0.04 (0.03–0.05)
36	12 (1.0)	yes	0.25 (0.19–0.35)
295	12 (1.0)	yes	0.26 (0.22–0.31)
81	9 (0.7)	yes	0.25 (0.17–0.36)
179	9 (0.7)	yes	0.41 (0.29–0.57)
10	6 (0.5)	yes	0.21 (0.16–0.29)
15	6 (0.5)	yes	0.17 (0.11–0.27)
248	6 (0.5)	yes	0.19 (0.15–0.25)
908	6 (0.5)	yes	0.11 (0.04–0.33)

**Table 2 antibiotics-12-00992-t002:** Performance of the prediction of penicillin G susceptibility, based on *penA* sequence.

Parameter	Value
Sensitivity	0.94
Specificity	0.97
Positive predictive value	0.96
Negative predictive value	0.96
Accuracy	0.96
Kapp coefficient	0.91

## Data Availability

The genomic data (FASTA files) for *N. meningitidis* can be retrieved from the PUBMLST.org site by filtering on country (France) and period (years 2017–2021).

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
