# Peer review of "Recent Evolution of Susceptibility to Beta-Lactams in Neisseria meningitidis"

_antibiotics, 2023, doi:10.3390/antibiotics12060992_

Round 1

Reviewer 1 Report

The authors have screened  different alleles for penA and have extensively the two most predominant ones, penA1 and penA9 in relation to beta-lactamase susceptibility.

Minor comments:

Line 175-176 : Please re write the statement and follow an uniform method to cite references 

Author Response

Comments and Suggestions for Authors

The authors have screened  different alleles for penA and have extensively the two most predominant ones, penA1 and penA9 in relation to beta-lactamase susceptibility.

Minor comments:

Line 175-176 : Please re write the statement and follow an uniform method to cite references 

Response of the authors: We apologize for that. The reference was corrected as requested by the reviewer (Please see line 178 of the revised version of the manuscript).

Reviewer 2 Report

The text is full of typos. Among the others, lines 82, 90, 140, 142, 159 twice, etc.

In the abstract, third-generation cephalosporins and the acquisition of ROB-1 beta-lactamase are mentioned. In the manuscript, neither is mentioned nor described.

Line 17. The authors declare they identified 74 different penA alleles. Line 114: The authors declare they detected 94 different penA alleles. The same is in line 178. Why?

Line 44. Full text of the PenI acronym should be added.

Amino acid residues are normally reported as an acronym of the amino acid followed by a number. In the text, the reviewer found a discrepancy. For instance, 310S, serine310, and S362. Please revise.

The full text of MIC is reported twice.

Line 94. The authors declare an increasing trend, with the highest values for 2021 with 58% of PenI isolates. In the abstract, the same data is 56%. Additionally, why did the authors mention Table 1?

In the abstract, the authors report significant data with no correlation along the manuscript. Data reported in the abstract are not supported by the analysis in the text. For instance, “Reduced susceptibility to 3GC remained limited at 2 %”, lines 16-17. Where did this topic discuss in the text?

The reviewer does not understand what lines 97-99 mean.

Line 188 = Which alleles? The same is in line 120.

Line 126 = acronym of GMT is not complete.

Line 147 = Is this a speculation? Data do not support the suggestion.

Line 175 = the reference is not properly reported.

Similar articles are reported in the literature, and susceptibility to antibiotics was assessed (for instance, 10.1093/infdis/jiac046). In order to robust the article in question, the susceptibility assessment should be performed.

The authors should discuss the impact of a specific mutation with respect to others. The only presence/absence of mutation in the 5 critical residues is not specific. The exact mutation should also be reported. Is one mutation more relevant than another one?

The reviewer found a lot of typos in the manuscript. Some sentences need to be rephrased.

Author Response

Comments and Suggestions for Authors

The text is full of typos. Among the others, lines 82, 90, 140, 142, 159 twice, etc.

Response of the authors: We are sorry for that and we corrected these typos.

In the abstract, third-generation cephalosporins and the acquisition of ROB-1 beta-lactamase are mentioned. In the manuscript, neither is mentioned nor described.

Response of the authors: We added data on both third-generation cephalosporins and the acquisition of ROB-1 beta-lactamase (Please refer to lines 96-99 and line 103 of the revised version of the manuscript).

Line 17. The authors declare they identified 74 different penA alleles. Line 114: The authors declare they detected 94 different penA alleles. The same is in line 178. Why?

Response of the authors: The correct number of penA alleles is 74 alleles. We corrected accordingly.

Line 44. Full text of the PenI acronym should be added.

Response of the authors: We added  PenI for penicillin intermediate phenotype (Please see line 44 of the revised version of the manuscript).

Amino acid residues are normally reported as an acronym of the amino acid followed by a number. In the text, the reviewer found a discrepancy. For instance, 310S, serine310, and S362. Please revise.

Response of the authors: We harmonized the nomenclature and S310 was used across the text.

The full text of MIC is reported twice.

Response of the authors: We only kept the first full text MIC (Please see line 44 of the revised version of the manuscript).

Line 94. The authors declare an increasing trend, with the highest values for 2021 with 58% of PenI isolates. In the abstract, the same data is 56%. Additionally, why did the authors mention Table 1?

Response of the authors: The correct percentage is 58%. We corrected this percentage in the abstract. We also deleted (Table 1) from line 94 of the revised version of the manuscript.

In the abstract, the authors report significant data with no correlation along the manuscript. Data reported in the abstract are not supported by the analysis in the text. For instance, “Reduced susceptibility to 3GC remained limited at 2 %”, lines 16-17. Where did this topic discuss in the text?

Response of the authors: We added data on both third-generation cephalosporins and the acquisition of ROB-1 beta-lactamase (Please refer to lines 96-99 and line 103 of the revised version of the manuscript).

The reviewer does not understand what lines 97-99 mean.

Response of the authors: The sentence was changed for more clarity.

Line 188 = Which alleles? The same is in line 120.

Response of the authors: the penS alleles for line were 1, 2, 3, 5, 22, 27 and 34  (see line 120 of the revised version of the manuscript).

Line 126 = acronym of GMT is not complete.

Response of the authors: The abbreviation "GMT" stands for geometric mean. To avoid any confusion with the terme "geometric mean titre" (mostly used for serological assays), we changed the abbreviation of GMT to GM in the MS.

Line 147 = Is this a speculation? Data do not support the suggestion.

Response of the authors: This sentence referred to what is suggested in reference 20 as indicated.

Line 175 = the reference is not properly reported.

Response of the authors: The reference is now properly indicated [7].

Similar articles are reported in the literature, and susceptibility to antibiotics was assessed (for instance, 10.1093/infdis/jiac046). In order to robust the article in question, the susceptibility assessment should be performed.

Response of the authors: Indeed, the antibiotic susceptibility testing was performed in our work for all the 1255 cultured isolates with similar conclusions as in  10.1093/infdis/jiac046). We also added this reference.

The authors should discuss the impact of a specific mutation with respect to others. The only presence/absence of mutation in the 5 critical residues is not specific. The exact mutation should also be reported. Is one mutation more relevant than another one?

Response of the authors: This issue was discussed in lines 203-206

Comments on the Quality of English Language

The reviewer found a lot of typos in the manuscript. Some sentences need to be rephrased.

Response of the authors: We reviewed the manuscript and corrected the typos and improved its English Language.

Reviewer 3 Report

The authors have chosen an extremely pertinent subject, as the COVID-19 and post-COVID scenario has resulted in the emergence of MDR strains, posing numerous difficulties for therapeutic interventions.

Beta-lactam antibiotics are a class of antibiotics that are widely used to treat bacterial infections, including pneumonia, sepsis, and meningitis, among others.

However, the overuse and misuse of Beta-lactam antibiotics have led to the emergence and spread of antibiotic-resistant bacteria, which are becoming increasingly difficult to treat. Antibiotic resistance occurs when bacteria develop the ability to withstand the effects of antibiotics, rendering them ineffective.

This is a serious problem because infections caused by antibiotic-resistant bacteria are more difficult to treat and can lead to longer hospital stays, higher healthcare costs, and increased mortality rates. In fact, it is estimated that by 2050, antibiotic-resistant infections could cause 10 million deaths annually, surpassing cancer as a leading cause of death.

Therefore, it is essential that we take action to address this issue. This includes reducing the unnecessary use of antibiotics, improving infection prevention and control measures, and developing new antibiotics and alternative treatments.

Antibiotic resistance is a serious threat to public health, and we must work together to combat it. By taking steps to preserve the effectiveness of Beta-lactam antibiotics, we can help ensure that these life-saving drugs remain effective for future generations.

I have few comments that need to be addressed:

1. The methodology section must be elaborated further with appropriate references so that an average reader can be able to understand.

2. Discussion: the significance of the central claims must be clarified in the context of the existing literature, what the present study adds to what was already done. And what gaps are the authors trying to fill-in? Add some recent references.

3. Before the conclusion, include the limitations and future implications of the study.

Author Response

Comments and Suggestions for Authors

 The authors have chosen an extremely pertinent subject, as the COVID-19 and post-COVID scenario has resulted in the emergence of MDR strains, posing numerous difficulties for therapeutic interventions.

Beta-lactam antibiotics are a class of antibiotics that are widely used to treat bacterial infections, including pneumonia, sepsis, and meningitis, among others.

However, the overuse and misuse of Beta-lactam antibiotics have led to the emergence and spread of antibiotic-resistant bacteria, which are becoming increasingly difficult to treat. Antibiotic resistance occurs when bacteria develop the ability to withstand the effects of antibiotics, rendering them ineffective.

This is a serious problem because infections caused by antibiotic-resistant bacteria are more difficult to treat and can lead to longer hospital stays, higher healthcare costs, and increased mortality rates. In fact, it is estimated that by 2050, antibiotic-resistant infections could cause 10 million deaths annually, surpassing cancer as a leading cause of death.

Therefore, it is essential that we take action to address this issue. This includes reducing the unnecessary use of antibiotics, improving infection prevention and control measures, and developing new antibiotics and alternative treatments.

Antibiotic resistance is a serious threat to public health, and we must work together to combat it. By taking steps to preserve the effectiveness of Beta-lactam antibiotics, we can help ensure that these life-saving drugs remain effective for future generations.

Response of the authors: We thank the reviewer for his comments to which we fully adhere.

I have few comments that need to be addressed:

  1. The methodology section must be elaborated further with appropriate references so that an average reader can be able to understand.

Response of the authors: We added more detailed information on the methodology and in particular on the antibiotic susceptibility testing (see line 231 to 248 in the revised version of the manuscript).

  1. Discussion: the significance of the central claims must be clarified in the context of the existing literature, what the present study adds to what was already done. And what gaps are the authors trying to fill-in? Add some recent references.

Response of the authors: We added a whole paragraph in the discussion on comparison of our data with those from the US and the UK with several agreement and added two addition references (21 and 22) (Please see lines 186-194 of the revised version of the manuscript).

  1. Before the conclusion, include the limitations and future implications of the study.

Response of the authors: We added limitations and future implications as requested by the Reviewer (please see lines 260-267 of the revised version of the manuscript).

Reviewer 4 Report

The paper Recent evolution of susceptibility to beta-Lactams in Neisseria meningitidis  by Deghmane et al. presents the data obtained as a result  well-planned research help to explain the reasons of reduced susceptibility to penicillin G of N. menngitidis (PenI) observed in the meningococcal disease cases. PenA alleles sequencing showed total correlation between the presence of the modification at the 5 critical positions of PBP2 protein and GMT of the MIC of penicillin G. The peer-reviewed manuscript brings in a lot of new information; one of most significant concern the possibility to design beta-lactams effective against PenI strains of N. meningitides and probably H. influenzae. These valuable results were obtained using UCSF Chimera, developed by the Resource for Biocomputing, Visualization, and Informatics at the University of California, San Francisco.

It would be interesting to compare the results of this paper with that reporting about the increase of PenI isolates in France, England, Wales and Northern Ireland since the early 2000s:

“Increase in penicillin-resistant invasive meningococcal serogroup W ST-11 complex isolates in England. Laura Willerton, Jay Lucidarme, Andrew Walker, Aiswarya Lekshmi, Stephen A. Clark, Steve J. Gray, Ray Borrow . Vaccine 39 (2021) 2719–2729”

However, there are a few minor points to consider for further improvements.

Minor comments

Title  Recent evolution of susceptibility to beta-Lactams in Neisseria meningitidis  ….  beta-lactams should be written in lower case

PenI: abbreviation need  to be defined = penicillin-susceptible, increased exposure

CI = confidence interval also is necessary to explain.

L 35 and 42 beta-lactams antibiotics or beta lactam antibiotics

L 36 their safety … safety  font color should be black and without space

L76 French national reference center for meningococci….  or  National Reference Center for the Meningococci

L 82 respectively) .. the dot is missing …. respectively).

L 90 minimal inhibitory (MIC) …. minimal inhibitory concentration (MIC)  see L 44

L 94 1mg/L …  space is missing …1 mg/L 

L 126 geometric means (GMT) …. Geometric Mean Titres (GMT)

L 223 recommendations[6, 7].. space is missing.. recommendations [6, 7].

L 224 and 225 described[23]. databases[24] . please insert spaces before parentheses

L 268 No References word before the list of publications

Author Response

Comments and Suggestions for Authors

The paper Recent evolution of susceptibility to beta-Lactams in Neisseria meningitidis  by Deghmane et al. presents the data obtained as a result  well-planned research help to explain the reasons of reduced susceptibility to penicillin G of N. menngitidis (PenI) observed in the meningococcal disease cases. PenA alleles sequencing showed total correlation between the presence of the modification at the 5 critical positions of PBP2 protein and GMT of the MIC of penicillin G. The peer-reviewed manuscript brings in a lot of new information; one of most significant concern the possibility to design beta-lactams effective against PenI strains of N. meningitides and probably H. influenzae. These valuable results were obtained using UCSF Chimera, developed by the Resource for Biocomputing, Visualization, and Informatics at the University of California, San Francisco.

It would be interesting to compare the results of this paper with that reporting about the increase of PenI isolates in France, England, Wales and Northern Ireland since the early 2000s:

“Increase in penicillin-resistant invasive meningococcal serogroup W ST-11 complex isolates in England. Laura Willerton, Jay Lucidarme, Andrew Walker, Aiswarya Lekshmi, Stephen A. Clark, Steve J. Gray, Ray Borrow . Vaccine 39 (2021) 2719–2729”

Response of the authors: The paper mentioned by the Reviewer focuses on isolates of serogroup W belonging to CC11 and comparison may not be relevant as our work analysed all serogroups. However, another paper from the same group may be more relevant  (Willerton, L.; Lucidarme, J.; Walker, A.; Lekshmi, A.; Clark, S. A.; Walsh, L.; Bai, X.; Lee-Jones, L.; Borrow, R., Antibiotic resistance among invasive Neisseria meningitidis isolates in England, Wales and Northern Ireland (2010/11 to 2018/19). PLoS One 2021, 16, (11), e0260677). We therefore added this paper for comparison (See please line of the revised version of the manuscript). We added a whole paragraph in the discussion on comparison of our data with those from the US and the UK with several agreement and added two addition references (21 and 22). Please see lines 186-194 of the revised version of the manuscript).

However, there are a few minor points to consider for further improvements.

Minor comments

Title  Recent evolution of susceptibility to beta-Lactams in Neisseria meningitidis  ….  beta-lactams should be written in lower case

Response of the authors: We're sorry for that. “beta-lactams” has been written in lower case in the title and has also been checked all over the MS

PenI: abbreviation need  to be defined = penicillin-susceptible, increased exposure

Response of the authors: The "PenI" abbreviation stands for "intermediate susceptibility" for penicillin G with MIC ≥ 0.125mg/L and ≤ 1mg/L. This has been defined in lines 43 and 44.

CI = confidence interval also is necessary to explain.

Response of the authors: The abbreviation CI (confidence interval) was defined. Please see line 108.

L 35 and 42 beta-lactams antibiotics or beta lactam antibiotics

Response of the authors:  “beta-lactams antibiotics” and  “beta lactam antibiotics” were corrected to beta-lactam antibiotics all over the text.

L 36 their safety … safety  font color should be black and without space

Response of the authors: The font color was corrected and the space removed.

L76 French national reference center for meningococci….  or  National Reference Center for the Meningococci

Response of the authors: National Reference Centre for Meningococci and Haemophilus influenzae (NRCMHi). Please see lines 76-77.

L 82 respectively) .. the dot is missing …. respectively).

Response of the authors:  The dot was inserted after the parenthesis.

L 90 minimal inhibitory (MIC) …. minimal inhibitory concentration (MIC)  see L 44

Response of the authors: As the abbreviation MIC was first defined in line 44, the sentence "minimal inhibitory concentration" was removed from the rest of the MS and only the abbreviation MIC was left including in the line 90.

L 94 1mg/L …  space is missing …1 mg/L 

Response of the authors: The space was inserted

L 126 geometric means (GMT) …. Geometric Mean Titres (GMT)

Response of the authors: The abbreviation "GMT" stands for geometric mean. To avoid any confusion with the terme "geometric mean titre" (mostly used for serological assays), we changed the abbreviation of GMT to GM in the MS.

L 223 recommendations[6, 7].. space is missing.. recommendations [6, 7].

Response of the authors: The space was inserted

L 224 and 225 described[23]. databases[24] . please insert spaces before parentheses

Response of the authors: Spaces were inserted between the words and the citations and checked for the whole text.

L 268 No References word before the list of publications

Response of the authors: The word “ References” was added.

Round 2

Reviewer 2 Report

The quality of the article was improved after the revision, and it deserves to be published in Antibiotics.